coping resources; functioning; mental health; refugee

**Corresponding author:**
Gulsah Kurt;
Email: g.kurt@unsw.edu.au

# Profiles of coping resources and their associations with mental health and social functioning among refugees in Indonesia

Gulsah Kurt[1,2] 📷, Philippa Specker[1,2], Belinda Liddell[1,3], David Keegan[4,5], Randy Nandyatama[6], Atika Yuanita[7], Rizka Argadianti Rachmah[1,7], Joel Hoffman[8], Shraddha Kashyap[9], Diah Tricesaria[10], Mitra Khakbaz[4], Zico Pestalozzi[7] and Angela Nickerson[1]

[1]School of Psychology, University of New South Wales, Sydney, Australia; [2]Australian Human Rights Institute, University of New South Wales, Sydney, Australia; [3]School of Psychological Sciences, University of Newcastle, Newcastle, Australia; [4]HOST International, Sydney, Australia; [5]School of Social Work, Excelsia University College, Macquarie Park, Sydney, Australia; [6]Department of International Relations, Universitas Gadjah Mada, Yogyakarta, Indonesia; [7]SUAKA, Indonesian Civil Society Association for Refugee Rights Protection, Jakarta, Indonesia; [8]School of Medicine, Faculty of Medicine and Health, The University of Sydney, Sydney, Australia; [9]Bilya Marlee School of Indigenous Studies, University of Western Australia, Perth, Australia and [10]School of Social Sciences, Monash University, Melbourne Australia

## Abstract

This study examined the role of coping resources – self-efficacy (problem-focused) and emotion regulation (emotion-focused) – in supporting mental health and social functioning among refugees in a transit setting in Indonesia. Using a latent profile analysis approach with 1,214 participants, three distinct coping profiles were identified: high coping resources, high emotion-focused coping resource, and low coping resources. Results showed that high coping resources were associated with better mental health and social functioning outcomes. Emotion-focused coping resources were more strongly associated with better mental health, while problem-focused coping resources were closely linked to social functioning. This study highlighted the importance of coping flexibility and offers practical implications for strength-based interventions in transit displacement settings.

## Impact statement

The majority of the world's refugees and asylum-seekers reside in low- and middle-income countries. Despite this, most evidence on refugee mental health disproportionately stems from studies conducted in high-income settings. Moreover, existing research has predominantly focused on symptoms and diagnoses of mental health disorders. There is an urgent need to systematically investigate resilience-related factors among refugees and to expand the focus beyond mental health outcomes to include broader aspects of functioning. Addressing these critical gaps, the present study examined coping resources and their associations with mental health and social functioning outcomes. Using a person-centered statistical approach in a large cohort study with refugees in Indonesia, we identified distinct subgroups based on key coping resources, uncovered several individual- and context-level predictors of group membership and explored associated mental health and social outcomes. Our findings provide initial empirical evidence for the potentially prominent role of emotion-focused coping in mental health, and of problem-focused coping in social functioning, while emphasizing that both resources are important for better outcomes. These results advance our understanding of the applicability of conventional stress-coping frameworks in the context of forced displacement and offer practical insights for designing effective, culturally responsive psychological interventions that foster both emotion- and problem-focused coping resources.

## Introduction

Global forced displacement has reached record numbers, with over 120 million people displaced. Most refugees and asylum seekers live in transit in low- and middle-income countries. Despite high exposure to a multitude of stressors, only one-third of refugees are likely to develop mental health problems at any point in their lives (Blackmore et al., 2020; Patanè et al., 2022). Yet, most research to date has focused on the risk factors for mental health and functional impairments, often emphasizing vulnerability and risk over strength and resilience. Given that resilience is a common response among refugees, it is important to shift the focus from a deficit-based

approach centered on the prevalence of mental disorders and the risk factors to a strengths-based approach that aims to understand the factors promoting and protecting psychological and social functioning among refugees (Nickerson et al., 2024). This provides critical information to guide the design and delivery of interventions that build on the existing resources and strategies utilized by refugees to foster long-term resilience and empowerment (Saleebey, 1996; Brun and Rapp, 2001).

To better understand resilience among refugees, it is crucial to examine coping processes through which refugees respond to and manage adversities. This process is key in the psychological and functional sequelae of exposure to stress and adversities (Taylor and Stanton, 2007). According to the transactional model of stress and coping (Lazarus and Folkman, 1984), coping is a dynamic and context-dependent process shaped by individuals' cognitive appraisal of stressors (e.g., threat or challenge) and their perceived coping resources (e.g., perceived ability to cope) to manage external or internal demands of the stressors. In this model, coping resources play a central role in shaping the coping process by influencing how individuals appraise the stressful situation and which strategies they adopt. Contemporary models of coping commonly use the distinction between problem-focused (exerting deliberate effort to manage the stressful situation itself) versus emotion-focused (regulating emotions arising from the stressful situation) coping (Lazarus and Folkman, 1984), both of which are implicated in mental health. Although earlier studies suggested the benefits of problem-focused coping over emotion-focused coping, accumulated evidence revealed the context-dependency of the effectiveness of problem-focused versus emotion-focused coping, known as "the goodness-of-fitness hypothesis" (Folkman and Moskowitz, 2004; Taylor and Stanton, 2007). Accordingly, several studies demonstrated the effectiveness of problem-focused coping in dealing with stress in controllable situations and emotion-focused coping in uncontrollable situations (Folkman and Moskowitz, 2004). Other studies highlighted the importance of coping flexibility – the adaptive use of both problem- and emotion-focused coping strategies – for better psychological adjustment, especially following traumatic experiences (Bonanno and Burton, 2013; Cheng et al., 2014; Heffer and Willoughby, 2017). Most of these studies have considered problem- and emotion-focused coping processes separately or compared their relative effectiveness. Little is known about naturally occurring profiles of individuals who might engage in distinct coping processes. For instance, some individuals may rely more on emotion-focused coping and less on problem-focused coping, or vice versa, while others may engage in both to a varying degree at the same time. Additionally, the existing evidence is largely focused on coping strategies with limited attention to coping resources in the coping process, despite their crucial role in shaping the availability and flexible use of problem- and emotion-focused strategies (Taylor and Stanton, 2007). Building on this mainstream coping distinction and extending it, we assert that coping resources, rather than coping strategies, provide a more comprehensive and adaptable foundation for understanding how refugees navigate and cope with stressors in transit settings. Given that personal coping resources are core psychological assets (Taylor and Stanton, 2007), in the present study, we focused on self-efficacy and emotion-regulation as supporting problem- and emotion-focused coping processes, respectively. This emphasis on coping resources reflects a shift from evaluating what refugees do to cope to a focus on what enables them to cope, contributing to a strengths-based understanding of refugee mental health.

Refugees in these settings often face protracted uncertainty, such as extended waiting for resettlement, and contend with stressors such as limited access to social services and legal rights (Nickerson et al., 2022b, 2023b). In the Asia-Pacific region, Indonesia is the main transit country currently hosting over 12,000 refugees, many of whom have been waiting for resettlement for over a decade (United Nations High Commissioner for Refugees, 2025). As a non-signatory to the 1951 Geneva Convention and its 1967 Protocol, Indonesia does not grant formal refugee status to those fleeing persecution and conflict. As such, asylum seekers must undergo a lengthy refugee status determination process through UNHCR, making Indonesia a particularly complex and uncertain setting (Curby, 2020). Beyond the assistance and support from international organizations and local initiatives, refugees[1] in Indonesia are legally restricted from formal employment and have limited access to health care and education, often living in a state of limbo for years (Brown, 2018; Amin, 2022). These conditions present a complex mix of both uncontrollable systemic conditions and those requiring both problem- and emotion-focused coping. Review studies from low-resource or transit settings also show that refugees often engage in broader categories of problem- and emotion-focused coping in response to varying degrees of stress (Seguin and Roberts, 2017; Posselt et al., 2019; Figueiredo and Petravičiūtė, 2025). However, consistent with broader trends in the coping literature, most of this evidence focuses on strategies rather than underlying coping resources and examines problem- and emotion-focused coping independently. Thus, understanding the distinct profiles of coping resources and how these profiles are related to certain adjustment outcomes is necessary to develop targeted, resource-based interventions to foster adaptive coping and resilience among refugees in transit settings.

Self-efficacy as a coping resource for problem-focused coping is defined as the belief in one's capacity to effectively manage stressful situations (Benight and Bandura, 2004). The perceived ability to manage stressors fosters reappraisals of stressors as manageable, thereby motivating the use of coping strategies that assist in managing or altering stressful situations, such as planning problem-solving strategies and seeking support (Taylor and Stanton, 2007; Groth et al., 2019; Van den Brande et al., 2020; Weigold et al., 2024). Prior research has shown that higher level of self-efficacy predicts better mental and physical health outcomes, such as lower levels of anxiety, depression, post-traumatic stress symptoms and somatic symptoms, among both general and trauma-affected populations (Luszczynska et al., 2009; Andersson et al., 2014; Gallagher et al., 2020). Similar findings were obtained from refugee studies, suggesting that self-efficacy is associated with positive mental health outcomes and resilience (Tip et al., 2020; Nickerson et al., 2022a; Pak et al., 2023).

Emotion regulation – one's ability to monitor, evaluate and modify emotional reactions in a specific situation (Gratz and Roemer, 2004) – is a key resource for the emotion-focused coping process. Studies have shown that emotion regulation difficulties are associated with an increased risk of experiencing psychological distress and functional impairments in diverse trauma-exposed individuals (Cloitre et al., 2019; Klemanski et al., 2012; Shepherd and Wild, 2014; Muñoz-Rivas et al., 2021). Similarly, studies with refugees showed that difficulties in emotion regulation are associated with increased symptoms of mental health disorders,

---

[1]We refer to all asylum-seeking individuals in Indonesia as refugees for the sake of consistency with the broader refugee mental health literature and to enhance readability.

displacement-related difficulties and impairments in social functioning in refugees (Nickerson et al., 2015; Koch et al., 2020; Specker et al., 2024a). There is also evidence that enhancing emotion regulation can help with the management of trauma-related stress among refugees (Nickerson et al., 2017). Taken together, these findings suggest that emotion regulation ability is an important resource to facilitate adaptive coping for refugees.

Despite substantial evidence on self-efficacy and emotion regulation as coping resources for refugees, studies to date have predominantly investigated these constructs separately, potentially overlooking how these may co-occur or interact to facilitate the coping process. Person-centered statistical approaches may overcome these limitations by elucidating patterns of responding across both types of resources, rather than viewing them in isolation or opposition (Ferguson et al., 2020). Accordingly, methods such as latent class/profile analysis have been increasingly implemented in recent years, including to examine profiles of displacement experiences and psychological outcomes among refugees (Nickerson et al., 2019; Sengoelge et al., 2019; Byrow et al., 2022). Yet, there is a paucity of research investigating coping resources among refugees and mental health and social functioning outcomes using this approach. Thus, the application of a person-centered approach in coping research reflects a methodological shift from isolating individual predictors to mapping profiles of coping resources in combination. This approach aligns well with the transactional model of stress and coping, which emphasizes the dynamic interplay between coping resources in shaping the coping process. This also enables us to capture a more realistic picture of the coping processes of refugees facing protracted uncertainty and continuous stressors in transit settings. In the present study, we aimed to examine: (1) distinct profiles of problem-focused (i.e., self-efficacy) and emotion-focused (i.e., emotion regulation) coping resources, (2) predictors of these coping profiles (e.g., demographics and conflict- and displacement-related factors) and (3) associated mental health and social functioning outcomes among refugees in Indonesia. Based on prior studies on coping typically yielded three to four profile resolutions (Doron et al., 2015; Nielsen and Knardahl, 2014; Kavčič et al., 2022), we deemed the emergence of four distinct profiles: higher on both resources, higher on self-efficacy, higher on emotion-regulation and lower on both. Drawing on coping flexibility, we hypothesized that the profile characterized by higher self-efficacy and emotion-regulation ability would demonstrate better mental health and social functioning outcomes than other profiles. The profile low on both types of resources would report worse outcomes. Considering the context-dependency of coping and the high degree of uncontrollable stressors in Indonesia, we predicted that a profile higher on emotion regulation would be associated with better mental health and social functioning than one higher on self-efficacy.

## Methods

### Participants and study design

The present study included the data from the first wave of an online longitudinal study conducted in Indonesia between 2020 and 2022. The recommended sample size for a person-centered analysis is between 500 and 1,000, based on the extant literature (Tein et al., 2013), so we aimed to sample at least 1,000 participants in the present study by considering both the required power and expected attrition rate in the longitudinal studies (Nickerson et al., 2023a). Due to the COVID-19 pandemic at the onset of the study, we opted for an online data collection to ensure the safety of participants and the research team. Participants were recruited via referrals from refugee services, community-based organizations in Jakarta and Bogor in Indonesia and social media. The inclusion criteria for this study were (1) being a refugee or asylum-seeker who arrived in Indonesia in 2013 or after, (2) being at least 18 years old and (3) being literate in one of the study languages (Arabic, Farsi, Dari, Somali or English). These language groups were purposively selected as they represent the majority of refugees in Indonesia (over 80%) (United Nations High Commissioner for Refugees, 2018). The self-administered online survey took approximately 1 hour. To mitigate barriers related to the online nature of the study, the survey was presented in a mobile-phone compatible format and available in the participant's language. The trained research staff were also available to support when needed. Participants were compensated with a grocery voucher of $USD 7 (IDR 100,000).

### Measures

All measures were translated and blind back-translated for each study language by the accredited translators. The translated measures were pilot-tested with refugee community members from different educational backgrounds to ensure comprehension, cultural relevance, and linguistic clarity.

### Coping resources

*Problem-focused coping resource.* We used the General Self-Efficacy Scale (Schwarzer and Jerusalem, 1995) to assess the problem-focused orientation among participants. They were asked to rate 10 items (e.g., "I can always manage to solve difficult problems if I try hard enough") on a 4-point Likert Scale (1 = not at all true, 4 = exactly true) to indicate how much each item applied to them. Higher scores indicate a greater tendency toward self-efficacy and thus, a problem-focused coping orientation. This scale has been used in prior studies with diverse groups of refugees (Nickerson et al., 2022a). Cronbach's alpha was 0.911 in this study.

*Emotion-focused coping resource.* Non-acceptance and impulse control subscales of the Difficulties in Emotion Regulation Scale (DERS; Gratz and Roemer, 2004) were used to measure participants' orientations toward regulating their emotions. Other subscales of the DERS were considered for inclusion in the emotion-focused coping resource, but omitted due to conceptual overlap with self-efficacy (e.g., as with the goals subscale) and mental health (e.g., as with the strategies subscale) or their focus on recognition or understanding emotions (as with the awareness and clarity subscales) preceding use of emotion-regulation strategies (Gratz and Roemer, 2004), rather than active regulation or management of emotions which is central to emotion-focused coping. We also aimed to capture specific dimensions of emotion regulation consistently linked to mental health (Short et al., 2016; Hallion et al., 2018). Six items (e.g., non-acceptance: "When I'm upset, I feel guilty for feeling that way" and impulse control: "When I'm upset, I become out of control") were rated on a 5-point Likert scale, ranging from "almost never (1)" to "almost always (5)." Items were reverse-coded so that higher scores on the scale indicate a greater orientation towards emotion-focused coping. Previous studies utilized this scale for use among refugee populations (Specker and Nickerson, 2019; Doolan et al., 2017; Koch et al., 2020; Liddell et al., 2023) and confirmed the factor structure (Specker et al., 2024a). The Cronbach's alpha was 0.837 for non-acceptance and 0.892 for impulse control.

### Mental health outcomes

*Depression symptoms.* Depression symptoms were measured using the Patient Health Questionnaire (PHQ-8) (Kroenke et al., 2009). Participants rated each item (e.g., feeling down, depressed, or hopeless) on a 4-point Likert scale (0 = not at all, 3 = nearly every day) to indicate how much they had been bothered by it in the past 2 weeks. This scale has been widely used in previous studies and has been validated in several languages, including Arabic, Farsi and Somali (Nallusamy et al., 2016; Sawaya et al., 2016; Dadfar et al., 2018). A mean score of the items was used in this study. Higher scores indicate more depressive symptoms (Cronbach's alpha = 0.885).

*Anxiety symptoms.* Anxiety symptoms were measured by the 7-item Generalized Anxiety Disorder Scale-7 (GAD-7) (Spitzer et al., 2006), in which participants indicated each item bothered them over the last 2 weeks (e.g., trouble relaxing) on a 4-point Likert scale (0 = not at all, 3 = nearly every day). This scale has been validated in several languages (Plummer et al., 2016) and used with refugee populations (Leiler et al., 2019). The mean score of the items was used in the study, with higher scores reflecting more anxiety symptoms (Cronbach's alpha = 0.926).

*Posttraumatic stress disorder symptoms.* An adapted version of the Posttraumatic Diagnostic Scale for DSM-IV (Foa et al., 1997) was used to measure the symptoms of post-traumatic stress disorder (PTSD). Four additional items (negative expectations about oneself/the world, distorted self or other blame, negative emotional states and reckless behavior) were added to the existing 16 items to reflect revised PTSD symptoms per the DSM-5. Participants reported how often each symptom (e.g., having upsetting thoughts/images about trauma) bothered them over the past month on a 4-point Likert scale (0 = not at all/only once, 3 = 5+ times a week/almost always). The items were averaged to indicate the level of PTSD symptoms experienced by the participants, with higher scores indicating more severe symptoms (Cronbach's alpha = 0.953).

### Social functioning outcomes

*Social engagement.* Three items from the Short Social Capital Assessment Tool (De Silva et al., 2007) were used to measure the level of social engagement among the participants. These items were the number of (1) groups (e.g., religious, sports, volunteer/charity) participants were active in, (2) groups providing emotional/economic support and (3) individuals providing emotional/economic support in the past 12 months. The total score of social engagement was calculated by summing these three items (Nickerson et al., 2022a), with a higher score indicating greater social engagement.

*Positive social support.* Positive social support was measured using 8 items developed by Araya et al. (2007). The items tap into different aspects of social support such as attachment, reassurance of worth, reliable alliance, and guidance (e.g., "There are people I can depend on to help me if I really need it") and are rated on a 5-point Likert scale ranging from 1 (strongly disagree) to 5 (strongly agree). Higher scores reflect greater perceived social support (Cronbach's alpha = 0.895).

### Conflict and displacement-related experiences

*Traumatic experiences.* The 16-item scale of The Harvard Trauma Questionnaire (Mollica et al., 1992) was used to measure the potentially traumatic experiences. Participants were asked to indicate if they experienced, witnessed or learned about each traumatic event (e.g., lack of food, imprisonment, torture, or serious injury) as yes (1) or no (0). The total number of traumatic events experienced or witnessed was calculated for use in this study. The scale has been widely used with refugee populations (Purgato et al., 2022).

*Post-displacement stressors.* A 42-item version of the Postmigration Living Difficulties Checklist (Steel et al., 1999) adapted to the Indonesian context was used to assess the wide range of social, economic, and legal stressors that the participants have been experiencing in the past 12 months. Each item is rated on a 5-point Likert Scale, ranging from 1 (was not a problem/did not happen) to 5 (a very serious problem). A mean score of the items was calculated to index the overall stressors encountered in the post-displacement setting.

### Data analysis

We conducted a latent profile analysis in Mplus Version 8.8 (Muthén and Muthén, 2023) following a multi-step approach (Ferguson et al., 2020). Using 16 continuous items for problem-focused orientation (10 items) and emotion-focused orientation (6 items), we first modeled a one-profile solution, followed by a sequential increase in the number of profiles modeled up to a 6-profile solution (Tein et al., 2013). Model fit was evaluated using several fit indices: lower Akaike's Information Criterion (AIC), Bayesian Information Criterion (BIC), Sample-size Adjusted Bayesian Information Criterion (SS-BIC), higher entropy, significant Vuong–Lo–Mendell–Rubin Likelihood Ratio Test (VLMR-LRT), Lo–Mendell–Rubin Likelihood Ratio Test (LMR-LRT) and theoretical conceptualization (Ferguson et al., 2020). Given the different rating scales of the indicator variables, we standardized the items to facilitate the interpretation. After selecting the best-fitting model, we assigned each participant to a specific profile based on their probabilities. In the second step, we used multi-nominal logistic regression to examine the predictors of the distinct profiles by including demographic variables such as age and gender, as well as pre- and post-displacement experiences, such as exposure to potentially traumatic events, post-displacement stressors in Indonesia, and length of stay in Indonesia. In the final step, we assessed whether the profiles were differentially associated with mental health and social functioning outcomes. To do so, we used the BCH method in Mplus, which uses observation weights reflecting the measurement error in the latent variable and thereby accounts for inaccuracy in profile classification (Asparouhov and Muthén 2014). Missing data on predictor variables were handled using multiple imputations, generating 20 random datasets. The imputed datasets were then used in the models with the predictors and outcomes of the latent profiles.

## Results

### Sample characteristics

The current sample consisted of 1,214 participants (862 male and 337 female) with a mean age of 30.59 (*SD* = 9.09). Participants' language backgrounds were Arabic (30.3%), Dari (21.1%), Farsi (18.3%), English (17.5%) and Somali (12.9%). Half of the sample were either married or in a relationship (45.6%) and had completed at least high school (49.5%). The average length of stay in Indonesia was 5.11 years (SD = 1.62).

**Table 1.** Model fit indices for latent profile models

| Models | AIC | BIC | SS-BIC | VLMR-LRT | LMR-LRT | Entropy |
|--------|-----|-----|--------|----------|---------|---------|
| 1. Profile model | 53,698.095 | 53,861.349 | 53,759.703 | | | |
| 2. Profiles model | 49,602.127 | 49,852.109 | 49,696.465 | 0 | 0 | 0.878 |
| 3. Profiles model | 47,789 | 48,125.711 | 47,916.068 | 0.0002 | 0.0002 | 0.888 |
| 4. Profiles model | 46,806.367 | 47,229.807 | 46,966.165 | 0.1076 | 0.1107 | 0.879 |
| 5. Profiles model | 45,912.292 | 46,422.46 | 46,104.819 | 0.0393 | 0.0403 | 0.899 |
| 6. Profiles model | 45,357.869 | 45,954.765 | 45,583.126 | 0.0283 | 0.0292 | 0.903 |

Abbreviations: AIC: Akaike's Information Criterion; BIC: Bayesian Information Criterion; SS-BIC: Sample-size Adjusted Bayesian Information Criterion; VLMR-LRT: Vuong-Lo–Mendell–Rubin Likelihood Ratio Test; LMR-LRT: Lo—Mendell–Rubin Likelihood Ratio Test.

### Latent profiles of coping resources

The model fit indices for the models from 1 to 6 profiles are given in Table 1. AIC, BIC, and SS-BIC values steadily improved from a 1-profile model to a 6-profile model. The LMR-LRT and VLMR-LRT indices suggested that the 3-profile model showed better fit than the four-profile model, as well as higher entropy. The VLMR-LRT and LMR-LRT suggested that the five and six-profile models also showed better fit than the four- and five-profile models, respectively, with entropy again increasing. Inspection of these models revealed that the five-profile model did not provide qualitatively distinguishable profiles, with some overlapping patterns across profiles despite the added model complexity. In addition to the conceptual overlap across the profiles, the six-profile model produced two profiles with small numbers of participants, one with 5% of participants and the other with less than 5% of participants. Thus, we retained the model with three profiles over models with five and six profiles considering the combination of clear differentiation of profiles and good model-fit indices. These three profiles can be described as profile 1, low on problem-focused and very low emotion-focused coping resources (low coping resources profile) ($N = 309$, 25.45%); profile 2, very low on problem-focused and high on emotion-focused coping resources (high emotion-focused coping resource profile) ($N = 315$, 25.95%); and profile 3, high on problem-focused and emotion-focused coping resources (high coping resources profile) ($N = 590$, 48.6%). The probabilities for the most likely profile membership for classes 1 to 3 were 0.945, 0.927 and 0.961, respectively. Table 2 presents the standardized item means for each class, as depicted in Figure 1.

### Predictors of coping resources profiles

Table 3 presents the predictors of latent profile membership. Gender and age did not significantly differ among the three profiles. Compared to the *low coping resources profile* (profile 1), participants in the *high emotion-focused coping resource profile* (profile 2) and the *high coping resources* profile (profile 3) had been in Indonesia for a longer time and reported lower levels of post-displacement stressors. Participants in the *high coping resources profile* (profile 3) reported more post-displacement stressors than those in the high emotion-focused coping resource profile (profile 2), and fewer traumatic experiences than those in the low coping resources profile (profile 1). Compared to the low coping resources profile, there were fewer Farsi- and Dari-speaking participants in the high-emotion coping resource and high coping resources profile. There were fewer English-speaking participants in the high coping resources profile than in the profiles of low coping resources and the high emotion-focused coping resources. Compared to the high emotion-focused coping resource profile, more Somali-speaking participants were in the high coping resource profiles compared to Arabic-speaking participants.

### Mental health and social outcomes of coping resources profiles

The results of the analysis for group differences between the three profiles on mental health and social functioning outcomes are presented in Table 4 and Figure 2. Compared to the high coping resources profile (profile 3), participants in the low coping resources (profile 1) and the high emotion-focused coping resource (profile 2) profiles reported significantly higher levels of depression ($\chi^2(2, 1{,}214) = 331.21$, $p = 0.000$, $\chi^2(2, 1{,}214) = 4.15$, $p = 0.042$, respectively) and anxiety symptoms ($\chi^2(2, 1{,}214) = 318.22$, $p = 0.000$, $\chi^2(2, 1{,}214) = 4.66$, $p = 0.031$, respectively). The low coping resources profile (profile 1) had more PTSD symptoms than the high emotion-focused coping resource (profile 2) ($\chi^2(2, 1{,}214) = 183.36$, $p = 0.000$) and high coping resources profiles (profile 3) ($\chi^2(2, 1{,}214) = 277.45$, $p = 0.000$). Those in the low coping resources profile (profile 1) had significantly higher levels of depression ($\chi^2(2, 1{,}214) = 165.77$, $p = 0.000$) and anxiety symptoms ($\chi^2(2, 1{,}214) = 170.70$, $p = 0.000$) than those in the high emotion-focused coping resource (profile 2).

For social functioning outcomes, participants in the high coping resources profile (profile 3), reported higher levels of social engagement and perceived positive social support than those in the high emotion-focused coping resource ($\chi^2(2, 1{,}214) = 11.64$, $p = 0.001$, $\chi^2(2, 1{,}214) = 92.19$, $p = 0.000$) and low coping resources profiles ($\chi^2(2, 1{,}214) = 4.86$, $p = 0.027$, $\chi^2(2, 1{,}214) = 83.35$, $p = 0.000$). There were no significant differences between those in the low coping resources (profile 1) and the high coping emotion-focused resource (profile 2) profiles in social engagement ($\chi^2(2, 1{,}214) = 1.26$, $p = 0.263$) and perceived positive social support ($\chi^2(2, 1{,}214) = 2.02$, $p = 0.155$).

Overall, these results showed the significant association of having both resources for mental health and social functioning outcomes, as well as differential associations of self-efficacy and emotion-regulation with these outcomes.

### Discussion

To our knowledge, this is the first study to investigate distinct groups of refugees based on their coping resources, specifically problem-focused (self-efficacy) and emotion-focused (emotion regulation), in a transit setting. Our results revealed three latent profiles of coping resources: those higher on both problem- and emotion-focused coping resources (high coping resources profile) (48.6%), those higher on emotion-focused, but very low on

**Table 2.** Standardized items mean by the latent profiles

| Items | Low coping resources profile | | | High emotion-focused coping resource profile | | | High coping resources profile | | |
|---|---|---|---|---|---|---|---|---|---|
| | Mean | SE | p | Mean | SE | p | Mean | SE | p |
| 1. I can always manage to solve difficult problems if I try hard enough. (SE) | −0.286 | 0.076 | 0 | −0.736 | 0.084 | 0 | 0.554 | 0.041 | 0 |
| 2. If someone opposes me, I can find means and ways to get what I want. (SE) | −0.136 | 0.066 | 0.04 | −0.642 | 0.081 | 0 | 0.426 | 0.047 | 0 |
| 3. It is easy for me to stick to my aims and accomplish my goals. (SE) | −0.254 | 0.076 | 0.001 | −0.77 | 0.08 | 0 | 0.557 | 0.043 | 0 |
| 4. I am confident that I could deal efficiently with unexpected events. (SE) | −0.233 | 0.075 | 0.002 | −0.765 | 0.084 | 0 | 0.539 | 0.044 | 0 |
| 5. Thanks to my resourcefulness, I know how to handle unforeseen situations. (SE) | −0.157 | 0.069 | 0.023 | −0.752 | 0.076 | 0 | 0.494 | 0.049 | 0 |
| 6. I can solve most problems if I invest the necessary effort. (SE) | −0.194 | 0.085 | 0.023 | −0.845 | 0.089 | 0 | 0.568 | 0.04 | 0 |
| 7. I can remain calm when facing difficulties because I can rely on my coping abilities. (SE) | −0.405 | 0.082 | 0 | −0.74 | 0.079 | 0 | 0.621 | 0.041 | 0 |
| 8. When I am confronted with a problem, I can usually find several solutions. (SE) | −0.313 | 0.085 | 0 | −0.801 | 0.086 | 0 | 0.605 | 0.041 | 0 |
| 9. If I am in trouble, I can usually think of something to do. (SE) | −0.364 | 0.086 | 0 | −0.701 | 0.09 | 0 | 0.572 | 0.037 | 0 |
| 10. No matter what comes my way, I'm usually able to handle it. (SE) | −0.281 | 0.08 | 0 | −0.764 | 0.083 | 0 | 0.561 | 0.044 | 0 |
| 11. When I am upset, I become embarrassed for feeling that way. (NA- ER) | −0.855 | 0.074 | 0 | 0.239 | 0.081 | 0.003 | 0.321 | 0.042 | 0 |
| 12. When I'm upset, I feel ashamed with myself for feeling that way. (NA-ER) | −0.876 | 0.068 | 0 | 0.291 | 0.088 | 0.001 | 0.303 | 0.044 | 0 |
| 13. When I'm upset, I feel guilty for feeling that way. (NA- ER) | −0.899 | 0.079 | 0 | 0.307 | 0.077 | 0 | 0.312 | 0.043 | 0 |
| 14. When I'm upset, I become out of control. (IC-ER) | −1.208 | 0.093 | 0 | 0.326 | 0.084 | 0 | 0.462 | 0.033 | 0 |
| 15. When I'm upset, I have difficulty controlling my behaviors. (IC-ER) | −1.23 | 0.1 | 0 | 0.335 | 0.083 | 0 | 0.473 | 0.032 | 0 |
| 16. When I'm upset, I lose control over my behaviors. (IC-ER) | −1.267 | 0.11 | 0 | 0.343 | 0.079 | 0 | 0.484 | 0.03 | 0 |

Abbreviations: SE: Self-efficacy; NA-ER: Non-acceptance subscale of difficulties in emotion regulation; IC-ER: Impulse control subscale of difficulties in emotion regulation.

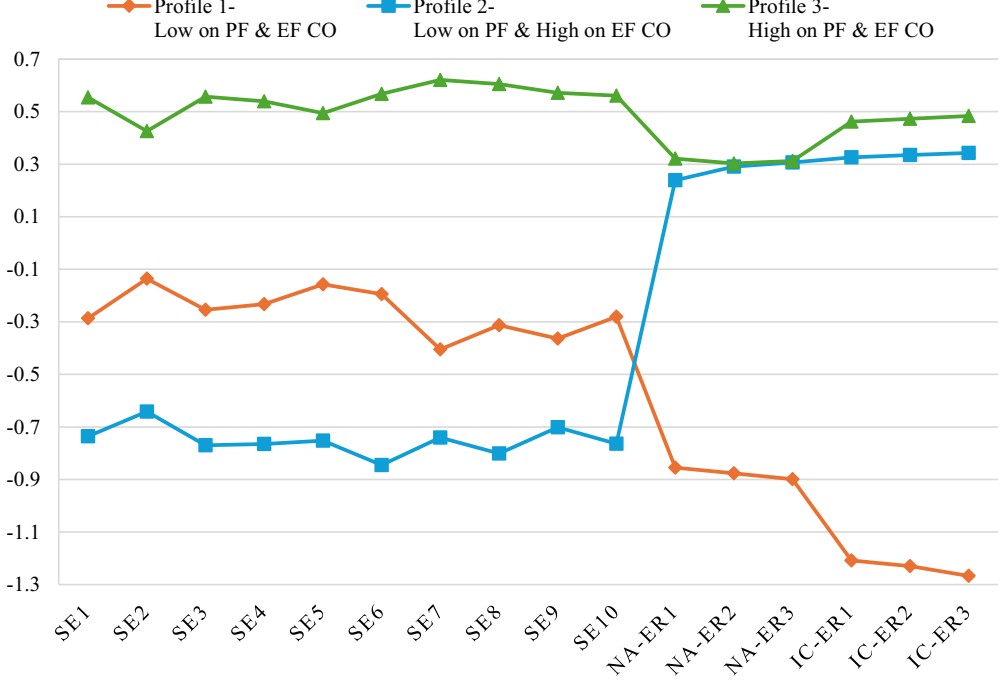

**Figure 1.** Means of latent profiles on problem-focused coping and emotion-focused coping.
*Note:* SE1-SE10 are rated on a 4-point Likert Scale (1 = not at all true to 4 = exactly true) while NA-ER1-IC-ER3 are rated on a 5-point Likert Scale (1 = almost always to 5 = almost never). Thus, the items were standardized to provide better comparability across the two scales.

**Table 3.** Predictors of latent profiles

|  | Estimate | SE | P |
|---|---|---|---|
| **Profile 1 versus profile 2** | | | |
| Age | −0.005 | 0.012 | 0.664 |
| Gender (ref = male) | 0.244 | 0.23 | 0.289 |
| Language (ref = Arabic) | | | |
| English | −0.346 | 0.297 | 0.244 |
| Farsi | −0.839 | 0.312 | 0.007 |
| Somali | −0.713 | 0.447 | 0.11 |
| Dari | −0.898 | 0.314 | 0.004 |
| Length of stay in Indonesia | 0.192 | 0.065 | 0.003 |
| Potentially traumatic experiences | −0.034 | 0.025 | 0.181 |
| Post-displacement stressors | −1.216 | 0.178 | 0 |
| **Profile 1 versus profile 3** | | | |
| Age | 0.01 | 0.011 | 0.347 |
| Gender | −0.09 | 0.201 | 0.654 |
| Language (ref = Arabic) | | | |
| English | −1.281 | 0.271 | 0 |
| Farsi | −1.325 | 0.261 | 0 |
| Somali | −0.113 | 0.368 | 0.758 |
| Dari | −0.927 | 0.256 | 0 |
| Length of stay in Indonesia | 0.214 | 0.056 | 0 |
| Potentially traumatic experiences | −0.057 | 0.021 | 0.007 |
| Post-displacement stressors | −0.837 | 0.16 | 0 |
| **Profile 2 versus profile 3** | | | |
| Age | 0.015 | 0.009 | 0.1 |
| Gender | −0.335 | 0.178 | 0.06 |
| Language (ref = Arabic) | | | |
| English | −0.935 | 0.243 | 0 |
| Farsi | −0.486 | 0.244 | 0.047 |
| Somali | 0.6 | 0.288 | 0.037 |
| Dari | −0.03 | 0.242 | 0.903 |
| Length of stay in Indonesia | 0.022 | 0.054 | 0.676 |
| Potentially traumatic experiences | −0.024 | 0.021 | 0.252 |
| Post-displacement stressors | 0.379 | 0.129 | 0.003 |

*Note:* Profile 1 = low coping resources, Profile 2 = high emotion-focused coping resource and Profile 3 = high coping resources.

problem-focused, resources (high emotion-focused coping resource profile) (25.9%) and those low on problem-focused and very low on emotion-focused coping resources (low coping resources profile) (25.45%).

As for mental health outcomes, those with high coping resources (both problem-focused and emotion-focused) reported fewer mental health problems, indicated by lower levels of PTSD, depression and anxiety symptoms compared to the other two profiles. It is notable that this profile comprised almost 50% of the sample, providing further evidence of high resilience among refugee populations. Participants in the high emotion-focused coping resource

profile also had significantly lower mental health problems than those low on both resources. These findings align with the existing coping literature, suggesting that better mental health is associated with the availability and utilization of diverse resources and strategies among individuals exposed to adversities, especially traumatic events, including refugees (Cheng et al., 2014; Heffer and Willoughby, 2017; Seguin and Roberts, 2017; Posselt et al., 2019; Figueiredo and Petravičiūtė, 2025). Consistent with the broader literature on coping flexibility (Bonanno et al., 2004; Galatzer-Levy et al., 2012; Bonanno and Burton, 2013; Specker et al., 2024b), our findings suggest that having diverse resources that allow the individual to move flexibly between different coping strategies can be advantageous for adapting to the demands of the post-displacement context in transit settings. This flexibility may then be associated with better mental health.

Although having both emotion- and problem-focused coping resources appears to be the most desirable in terms of mental health in the forced displacement context, it may not always be possible for refugees to have high levels of both. One notable finding of this study was that participants high on emotion-focused coping resources had fewer mental health symptoms than participants in the low resources profile (low problem-focused and very low emotion-focused resource). As both of these profiles exhibited low levels of a problem-focused resource, the main distinction between these two groups was the level of emotion-focused coping resources. This suggests that emotion-focused coping resources may play a more salient role in predicting better mental health outcomes when resources are limited. This aligns with the goodness-of-fit hypothesis suggesting that emotion-focused coping can provide a relative advantage over problem-focused coping in uncontrollable stressful circumstances (Folkman and Moskowitz, 2004).

Regarding social functioning outcomes (as indexed by social engagement and social support), we found a similar pattern of results to that of mental health outcomes. Specifically, we found that those higher on both types of resources reported higher levels of social engagement and perceived positive social support than the other two profiles. Unlike mental health outcomes, there was no significant difference between those in the higher emotion-focused profile and those in the low resources profile on any of the social functioning outcomes. This finding suggests that emotion-focused coping resource alone was not sufficient to improve social functioning outcomes in our sample. The fact that the main difference between the profiles of high coping resources and high emotion-focused coping resource lies in the problem-focused resource might indicate the prominent role of problem-focused coping resources in social functioning. Effective social functioning in a new context might require an overt action to successfully navigate contextual, cultural and linguistic challenges and form new relationships (Ryan et al., 2008), which are central to problem-focused coping. Nonetheless, it is important to note that both the effectiveness and accessibility of coping resources depend on the context, shaped by the broader sociocultural factors (Figueiredo and Petravičiūtė, 2025). For instance, in a setting with relatively greater access to basic services, emotion-focused coping might be sufficient to support both well-being and social functioning among refugees (Kurt et al., 2021). Furthermore, the directionality of the relationship between coping resources and social engagement cannot be discerned as social engagement might have contributed to the accumulation of the coping resources. For instance, participating in community activities such as cultural events or volunteer activities might provide opportunities to expand social networks, help

**Table 4.** Multiple group differences between latent profiles on mental health and social functioning outcomes

| Outcomes | Profile 1-Low Coping Resources | Profile 2-High Emotion-Focused Coping Resource | Profile 3-High Coping Resources | Group differences between profiles |
|---|---|---|---|---|
| *Mental health outcomes* | M (SE) | M (SE) | M (SE) | |
| Depressive symptoms | 1.981 (0.038) | 1.175 (0.047) | 1.053 (0.033) | 1 > 2 > 3 |
| Anxiety symptoms | 2.119 (0.052) | 1.086 (0.056) | 0.931 (0.041) | 1 > 2 > 3 |
| PTSD symptoms | 1.654 (0.046) | 0.770 (0.044) | 0.726 (0.031) | 1 > 2 = 3 |
| *Social functioning outcomes* | M (SE) | M (SE) | M (SE) | |
| Social Engagement | 1.891 (0.138) | 1.669 (0.132) | 2.303 (0.122) | 3 > 1 = 2 |
| Positive Social Support | 3.120 (0.047) | 3.014 (0.055) | 3.656 (0.035) | 3 > 1 = 2 |

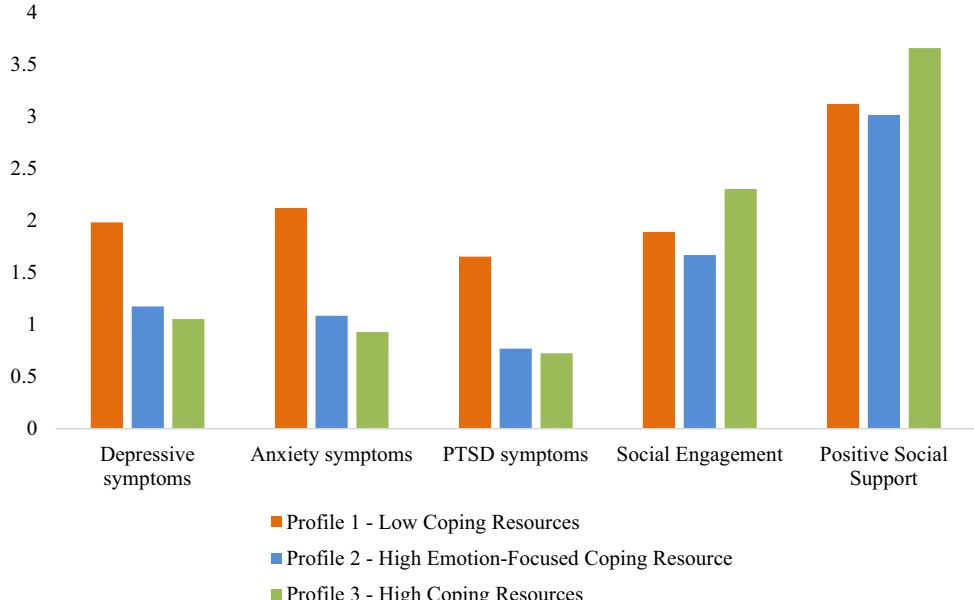

**Figure 2.** Mean differences across three profiles on mental health and social functioning outcomes.

navigate challenges and foster self-efficacy in managing contextual challenges.

We identified several predictors of coping resource profiles. First, language appeared as a significant predictor of profile membership. Compared to Arabic-speaking refugees, there were fewer Farsi- and Dari-speaking participants in the high emotion-focused resource profile and the high resources profile compared to the low resources profile. Similarly, fewer English-speaking participants were in the high coping resources profile than the other two profiles. Conversely, more Somali-speaking participants were in the high coping resources profiles compared to Arabic-speaking participants than the high emotion-focused coping resource profile. These findings highlight potential cultural differences in coping resources. While the coping process is mostly studied considering the demands and controllability of circumstances, these differences direct attention to how cultural/linguistic differences might determine the availability and utilization of coping resources, potentially through the differences in community support and culturally specific coping practices (Chun et al., 2006). Furthermore, participants in the high coping resources and high emotion-focused coping

resource profiles reported significantly longer time in Indonesia than those in the low resources profile. Staying in Indonesia for a longer time might have led individuals to find ways to develop or access coping resources. Further studies investigating changes in coping resources over time might clarify how these resources change over time. Those in the low coping resources profile also reported experiencing the most post-displacement stressors, followed by those in the high coping resources profile. As for potentially traumatic experiences, the low coping resources profile reported a significantly higher number of traumatic experiences than the high coping resources profile. These findings show that experiencing more traumatic experiences and displacement stressors is likely to diminish the availability of coping resources at higher levels (Steel et al., 2009; Hou et al., 2020).

The current findings should be interpreted with some limitations. First, while we chose self-efficacy and emotion-regulation as resources supporting problem- and emotion-focused coping processes, respectively, based on empirical evidence and theoretical considerations, coping resources are not limited to these. Future studies should consider a wider range of coping resources to

advance our understanding of protective and promotive factors for refugees. Furthermore, it is important to examine how cultural norms and religious beliefs shape the coping process through influencing the perception of stressors, the acceptability of certain coping strategies and use of community-based support mechanisms such as spiritual support (Seguin and Roberts, 2017). Such studies can be complemented by qualitative investigations, allowing community members to report on culturally specific coping resources and shed light on the differences observed among language groups. Second, although online data collection is a viable method to collect data from hard-to-reach populations (Bonevski et al., 2014), this might have led to a biased sample of those with higher levels of digital literacy and education. Thus, the current findings may not be broadly generalizable to the overall refugee population in Indonesia and other transit settings. This limitation is particularly relevant given that Rohingya refugees were not included in the study due to literacy-related barriers. It is important for future studies to address this, as Rohingya refugees have become a growing group of asylum seekers in Indonesia, especially since 2023. Furthermore, we measured post-displacement stressors using self-report which might reflect the level of psychological distress associated with each stressor rather than the objective stressors. We also used a composite score of the overall stressors, not allowing us to inspect the role of each unique stressor. The items of emotion-regulation were negatively worded, while self-efficacy items were positively worded. Although reverse-coded the emotion regulation items, valence differences may have influenced responses. Finally, the current findings are specific to some refugees in Indonesia during the study period; therefore, future studies should validate the emergence of the three coping resource profiles with other refugee groups in Indonesia and in other transit contexts.

The present study has several theoretical and clinical implications. First, the current findings provide initial evidence for the mental health and social functioning benefits of having both emotion- and problem-focused coping resources that might potentially promote coping flexibility among refugees in transit settings. So far, the majority of coping studies have included non-refugee, high-income populations and focused on health-related outcomes. Thus, the present study extends the application of coping frameworks to refugees in protracted transit settings and goes beyond health-related outcomes, including social functioning. These findings also suggest a further refinement of the transactional model of stress and coping, considering the idiosyncratic nature of a prolonged forced displacement. In this vein, identification of distinct resources profiles supports that the traditional problem versus emotion-focused coping divide may not fully capture the dynamic and context-dependent nature of coping, particularly in highly constrained settings. Furthermore, our findings indicate that mental health and social functioning are likely distinct yet related constructs. This might reflect the multi-faceted nature of the concept of well-being which includes various dimensions such as psychological, physical and social aspects (Jarden and Roache, 2023). Although the high-resources profile had overall better mental health and social functioning outcomes, the differences between the high emotion-focused and low resources profiles hinted to us that key determinants of mental health and social functioning might differ. Future studies could investigate distinct coping processes of mental health and social functioning and their underlying mechanisms to inform the development of targeted interventions.

The fact that the low-resources group reported the most displacement stressors underscores the critical need and urgency for political and structural reforms to address barriers to mental health and social functioning. In highly uncontrollable, prolonged transit settings, there is a limited capacity of individuals to change their circumstances, and individual coping efforts may not be sufficient. Furthermore, given the current conditions in Indonesia, especially as highlighted by the experiences of recent Rohingya arrivals (Hilmansyah, 2025), refugees' abilities to access, develop, and accumulate resources are extremely limited. Thus, it is imperative for policies to provide supportive environments that enable refugees to cultivate agency and efficacy within their settings. This can be achieved by increasing access to health services, education, and employment opportunities and implementing inclusive integration strategies conducive to social interactions with the host community (Hynie, 2018).

As for clinical implications, the findings highlight the potential importance of strengthening both problem-focused and emotion-focused coping resources to promote overall well-being among refugees. Culturally adapted, scalable interventions to foster emotion and problem-focused coping resources can help alleviate mental health problems and support social functioning (McDermott et al., 2024). Similarly, resource-based, resilience-oriented interventions that focus on the identification of strengths and enhancing resources have been found effective in promoting and protecting overall well-being among refugees, particularly in contexts where access to external support is limited (Ciaramella et al., 2022; de Alpuim-Gonçalves et al., 2025). Considering limited access to health care and psychosocial support among refugees in Indonesia, such interventions may take the form of community-based psychological programs in which refugee community members actively and meaningfully contribute to both design and implementation to facilitate empowerment and promote a sense of ownership among the community. This participatory approach can help address some shortcomings in traditional healthcare practices, such as limited cultural adaptations and ad-hoc community involvement by enhancing the acceptability, feasibility and sustainability of these interventions for refugees (Wallerstein and Duran 2010; Riza et al., 2020). Recent scoping reviews showed that training programs for healthcare workers supporting refugees should prioritize a strength-based approach, including understanding of cultural practices and inclusion of community members to offer effective and sustainable care to refugee populations (Riza et al., 2020; de Alpuim-Gonçalves et al., 2025). A recent example of a co-designed, resource-based psychological intervention demonstrated the feasibility and acceptability of such an approach, with significant improvements in well-being among refugee women (Greene et al., 2023).

To conclude, the present study provided the first empirical evidence on coping resources among refugees using a person-centered approach and associated mental health and social functioning outcomes. The key findings highlighted the importance of having higher levels of problem- and emotion-focused coping resources for better mental health and social functioning outcomes. Future research should investigate the longitudinal links between changes in coping resources and outcomes. Overall, this study offers important insights into the development of culturally tailored psychosocial interventions targeting both types of coping resources, especially self-efficacy and emotion-regulation, to improve the overall well-being of refugees.

**Open peer review.** To view the open peer review materials for this article, please visit http://doi.org/10.1017/gmh.2025.10053.

**Data availability statement.** The data are available upon reasonable request made to the corresponding author.

**Acknowledgements.** We would like to express our gratitude to the study participants and acknowledge the contributions of our study partners, HOST International, SUAKA, and Universitas Gadjah Mada. Finally, we would like to thank Yunizar Adiputera and Shaila Tieken.

**Authors contribution.** **G.K.:** Conceptualization, Formal analysis, Writing-Original Draft; **P.S:** Data Curation, Writing- Review & Editing; **B.L, D.K., R.N. & A.Y.:** Conceptualization, Methodology, Funding Acquisition, Writing-Review & Editing; **J.H. & S.K.**: Methodology, Project Administration, Data Curation, Writing- Review & Editing; **R.A.R, D.T, M.K, & Z.P.:** Project Administration, Investigation, Writing- Review & Editing. **A.N.:** Conceptualization, Methodology, Funding Acquisition, Supervision, Writing- Review & Editing. All authors critically reviewed and approved the final version of the manuscript.

**Financial support.** The current study was supported by an Australian Research Council Linkage Grant (LP170100852). P.S. was supported by an MQ: Transforming Mental Health Postdoctoral Scholarship (MPSIP\15). A.N. (2018104) was supported by an Australian National Health and Medical Research Council Investigator Leadership Grant.

**Competing interests.** The authors declare no competing interests.

**Ethics approval.** The study was approved by the UNSW Human Research Ethics Committee (HC190494).

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
