## [Reviewer Report]

TITLE AND ABSTRACT

Title Evaluation:

The title of the current article is informative, but it is rather long and could be simplified to make it more interesting and memorable. The phrase ‘A person-centred approach’ does reflect the distinctive methodology, but its position at the end of the title weakens the strength of the main topic.

Title Suggestion:

Understanding Coping Resources in Mental Health and Social Functioning of Refugees in Indonesia: A Profile-Based Approach

Abstract Evaluation:

The abstract is generally concise and well explains the background, methods, results, and implications. However, technical information such as profile percentages could be moved to the results section, and the narrative could be simplified to be more communicative for interdisciplinary readers.

Abstract Suggestions:

This study explored the role of coping resources (problem-focused and emotion-focused) on mental health and social functioning among refugees in Indonesia. Three distinct coping profiles were found using a latent profile analysis approach with 1,214 participants. Results showed that a combination of coping resources provided the best mental and social outcomes. This study highlights the importance of coping flexibility and practical implications in contextualised interventions in transit refugee settings.

INTRODUCTION

Evaluation:

The introduction presents the relevant global context, raises essential research gaps, and mentions theoretical contributions. However, there is limited discussion of the cultural context of Indonesia as the study site.

Suggestion:

The introduction is recommended to strengthen the integration of the local Indonesian context and highlight the importance of cultural adaptation in coping strategies. This will increase the value of this study’s contribution globally and locally.

METHODS

Evaluation:

The research methods were detailed and utilised a sophisticated statistical approach (LPA), with validation of measurement tools and online data collection appropriate for hard-to-reach populations. However, online methods' statistical power and possible bias were not explicitly explained.

Suggestion:

- Add an explanation of sample size justification and statistical power tests.

- Discuss potential bias due to online participant selection and limitations in population representation.

- Propose a future longitudinal design to look at the dynamics of changes in coping resources.

RESULTS

Evaluation:

Results were presented with clarity and depth, both descriptively and inferentially. The discovery of three distinct coping profiles enriches the literature on coping flexibility and refugee mental health.

Reasons for Strength:

- The measurement tool has been validated and widely used in refugee populations.

- Statistical findings are significant, and the LPA approach provides a rich picture of differentiation.

- Practical findings that can be used by humanitarian organisations, psychologists, and policymakers.

Suggestions:

- Visualisation of the profile tables and diagrams could be made more intuitive by highlighting key outcomes.

- Include the direct practical implications of each profile in the results to clarify its contribution.

DISCUSSION

Evaluation:

The discussion is well structured, compares the results with the literature, and transparently presents the study’s limitations. However, cross-cultural relevance and future research directions need to be emphasised.

Suggestion:

- Affirm the contribution of this study to coping models in the context of global displacement, particularly in transit countries.

- Add comparisons with similar studies in other regions (e.g. Europe or Africa).

- Discuss further how these results can be implemented in concrete interventions.

CONCLUSIONS

Evaluation:

The conclusions reflect the findings well but are still generalised.

Suggestion:

Add:

- Practical Implications: ‘Improving coping resources, especially in terms of emotion regulation and self-efficacy, should be a focus of psychosocial programmes for refugees.’

- Cultural Relevance: ‘Culturally tailored interventions are needed to increase the effectiveness of coping strategies in complex refugee settings.’

- Future Research Directions: ‘Longitudinal research to observe the dynamics of coping and social adaptation over the long term.’

ORIGINALITY AND ETHICS

Evaluation:

There is no indication of plagiarism or duplication of studies by the authors. Conflict of interest declaration and ethical approval have been correctly submitted.

---

## [Reviewer Report]

REVIEW

Investigating the role of coping resources in mental health and social functioning among refugees in Indonesia: A person-centered approach

Introduction

1. Problems: The introduction clearly identifies the global issue of forced displacement, but does not provide specific data or statistics on Indonesia, which would strengthen the relevance of the study in the local context. Recommendation: Add more specific data or statistics regarding the number of displaced people in Indonesia and the challenges they face to provide a more concrete picture of the relevance of this study in Indonesia.

2. Problems: The discussion of the concepts of self-efficacy and emotion regulation in Lazarus and Folkman’s (1984) theory is less clear, so the relationship with the study objectives is not fully explained. Recommendation: Expand the discussion of the theoretical basis underlying these concepts. Explain how Lazarus and Folkman’s (1984) theories of stress and coping inform the study, and relate them to the research objectives more clearly.

3. Problems: The introduction has cited some key literature, but does not include recent literature relevant to this topic, particularly in relation to refugees in Indonesia. Recommendation: Expand the literature review by adding references from relevant recent research on coping resources in the context of Indonesia or refugees in developing countries, to enrich the context of this research.

Method

4. For respondents, why did you not include Rohingya refugees in Indonesia, especially in Aceh, which is a problem in Indonesia?

5. Problems: This study relied on Latent Profile Analysis (LPA) to identify coping profiles, but did not provide external verification or replication to ensure that the profiles found remained consistent across different contexts or times. This reduces the potential for the findings to be applied beyond the sample used. Suggestion: To increase the validity and generalizability of the results, the authors should test these LPA findings using additional data or conduct replications across different refugee contexts or longitudinally. In addition, the LPA results should be verified using other methods such as cluster analysis to compare the consistency of the coping profiles found.

6. Issue: Although the authors used well-established instruments such as the General Self-Efficacy Scale (GSES) and Difficulties in Emotion Regulation Scale (DERS), there was no in-depth discussion of their validity and reliability in the context of refugee populations, who may have unique challenges in responding to these questions. Suggestion: The authors should include a discussion of how the instruments used were tested for validity and reliability in refugee populations, including relevant psychometric analyses. Local validation for the refugee context in Indonesia would strengthen the credibility of the results. A comprehensive explanation of which language the instrument was distributed in is needed, given the different backgrounds of refugees in terms of participant characteristics.

Result

7. In the respondent characteristics section, it is necessary to display the existing refugee categories.

8. Improved Visualization: Including images or graphs to visually depict the distribution of coping resources across profiles and their impact on mental health and social outcomes will improve reader comprehension. For example, a bar graph or scatter diagram showing the relationship between coping resources and mental health outcomes (e.g., depression, anxiety) across the three profiles would make the data easier to understand.

Discussion

9. Issues: Although the discussion addresses the findings in the context of previous literature, most of the references used appear to be from older studies or based on non-refugee populations. This may limit the relevance of the findings in the evolving context of refugee research. Suggestion: The authors need to integrate more references from recent literature, especially with regard to refugee research and coping resource-based interventions. This will strengthen the contribution of this research in a more modern and relevant context. Research focusing on resilience-based and psychosocial interventions for refugees should be prioritized.

10. The authors should delve deeper into the mechanisms that explain how emotion-focused coping can affect mental health and problem-focused coping can improve social functioning. A more detailed explanation of how these copings interact with other factors, such as social support or cultural context, would provide deeper and more contextualized insights.

11. The discussion does not go far enough in considering how broader contextual factors (such as state policies, access to health services, and social support within the community) affect the utilization of coping resources. Refugees in Indonesia are often trapped in conditions of long-term uncertainty, which affects their ability to access or develop coping resources. Some of the surface cases of refugees in Indonesia are Rohingya who have been rejected by the people of Aceh, some research has already studied it, the author needs to include it.

12. The authors should provide more details on how resilience-based interventions can be designed and implemented in the refugee context in Indonesia. This includes recommendations for training programs for social workers or psychologists working with refugees, as well as how community-based interventions can be facilitated.

13. The authors need to add a clearer discussion of methodological limitations, such as potential sample bias (especially in online data collection) and how limited access to certain refugee groups (e.g., those without internet access) might affect the findings. In addition, it is important to mention the limitations of using cross-sectional data that may not allow for a deeper understanding of changes in coping over time.

---

## [Reviewer Report]

Thank you for the submitted manuscript. In general, this paper presents interesting findings regarding five coping profiles in refugees in Indonesia using the latent profile analysis approach. However, several things need to be considered to strengthen the academic quality of this paper.

The background is still too dominated by quotations from previous literature without any elaboration or clear position from the author himself. The tendency to summarise other experts' opinions without providing original arguments makes the background narrative feel descriptive and less analytical. I suggest that the authors not only cite theories or previous findings, but also start building their own arguments regarding the urgency and relevance of this study. In other words, the author also needs to strengthen the rationale of the study by explicitly conveying the theoretical position or contribution offered.

The methods section has presented sufficient basic information regarding the longitudinal study design, participant recruitment procedures, and the instruments used to measure coping resources. However, there are several aspects that need to be clarified to improve the transparency and replicability of this study. Firstly, although the study is said to use a person-centred approach, it is not explained in detail what type of approach (e.g. Latent Profile Analysis or other methods) and how the analysis procedures were carried out. This explanation is important for readers to understand how the classification of coping profiles was formed. Secondly, online data collection methods are relevant in the context of a pandemic, but it is not explained how the digital divide is overcome, or how the validity of online data is maintained. Third, there is no mention of triangulation or internal validation measures to ensure measurement accuracy, which is important in cross-cultural studies involving respondents with different language and cultural backgrounds. Finally, it would be better if the authors also linked the chosen methodological approach with the underlying theoretical framework, so that there is consistency between the objectives, methods and data interpretation.

This article’s discussion has presented interpretations of the findings systematically, including attempts to link with previous literature, particularly theories of coping and refugee vulnerability. However, the depth of theoretical analysis still needs to be improved. The authors seem to tend to compare the findings with previous studies descriptively, without really establishing a critical dialogue with the theory or showing how the results of this study extend, revise, or even challenge existing theories. For example, the five coping profiles found actually open up space to reevaluate the classic separation between problem-focused and emotion-focused coping in the context of long-term displacement, but this opportunity has not been fully utilised in the discussion. In addition, while the local context of Indonesia is mentioned, the discussion is not exploratory enough in highlighting the unique dynamics of a transit country like Indonesia - particularly structural limitations and their impact on coping processes. In other words, the authors have not fully demonstrated the theoretical and contextual contributions of this study, which could be its main strength if developed further.

---

## [Editor Report]

All three reviewers comment the paper’s contribution to understanding coping resources among refugees in Indonesia using a latent profile analysis and person-centered approach. The findings are seen as novel and relevant, particularly in the under-researched context of a transit country. However, significant revisions are needed to improve conceptual and methodological clarity. Key areas for the authors to consider include strengthening the introduction by incorporating data from Indonesia and stronger theoretical framing, clarifying methodological procedures and addressing sampling limitations, and improving the discussion by deepening engagement with coping theories/frameworks and the specific challenges of transit contexts. We hope the authors will consider revising this manuscript according to these reviewers' suggestions.

---

## [Reviewer Report]

Comments to the Author

Thank you for your thoughtful and significant improvements to the manuscript titled “Profiles of Coping Resources and Their Associations with Mental Health and Social Functioning among Refugees in Indonesia.”

Minor Issues to Address:

Cultural Specificity in Discussion: Although the context of Indonesia is well-described, the Discussion could benefit from a more direct engagement with how cultural norms or local coping mechanisms (e.g., community-based spiritual support) might influence coping strategies.

Visual Clarity in Tables/Figures: Some tables and diagrams, especially those describing latent profiles, could be made more intuitive for broader readers by highlighting key findings or using color-coded patterns (if permitted by journal style).

Abstract Refinement: While improved, the abstract could be further tightened. Avoid using technical terms such as “salient” or “pronounced” that may not be reader-friendly to interdisciplinary audiences.

---

## [Editor Report]

Thank you for your comprehensive revisions to the manuscript. The reviewer was generally very positive about the improvements and now only suggests a few minor revisions. Please focus on their suggestions to provide some additional context to the discussion and editing the abstract to appeal to a broader audience.

---

## [Reviewer Report]

Comments to the Author

Strengths:

1. This article is methodologically robust and contributes significantly to the literature on mental health and social adaptation among refugees, particularly in middle-income transit countries such as Indonesia.

2. Using latent profile analysis on a large sample (n=1,214) strengthens the validity of the findings and enriches our understanding of natural coping profiles among refugees.

3. The findings on the critical role of coping flexibility, with differentiation between emotion-focused and problem-focused coping, offer relevant practical implications for developing strength-based interventions in refugee settings.

4. Explaining the Indonesian context and structural limitations is informative and adds depth to the discussion.

Areas for Minor Revision:

1. Justification for Construct Selection: Although the explanation regarding the selection of self-efficacy and emotion regulation is already quite good, the authors are advised to add one or two sentences that explicitly highlight why these two constructs were selected and why other constructs were not included, to convince readers from different disciplines better.

2. Representation of Minority Groups: The author mentions that the Rohingya group was not involved in this study. It would be better if this limitation were also emphasised in the conclusion, and the author suggests further exploration that is more inclusive of refugee groups that are highly vulnerable in terms of literacy or digital literacy.

3. Practical Implications: Although there is already an emphasis on community-based interventions, it would be stronger if the author added one concrete example (e.g., a specific intervention implemented in Indonesia or another transit country) as a brief reference.

4. English: Overall, the language is perfect. However, some sentences in the discussion section are long and could be simplified to make them easier for non-native readers.

---

## [Editor Report]

Thanks to the authors for submitting this revised version of your manuscript. The revised version is methodologically robust, well-written, and makes a valuable contribution to the literature on mental health and social functioning among refugees.

The reviewer commended the strength of your study design, including the use of latent profile analysis, as well as the clarity and relevance of your findings. In particular, the differentiation of coping strategies and the attention to context-specific structural limitations provide important insights with practical implications for intervention development.

The reviewer offered several constructive suggestions regarding justification of construct selection, representation of vulnerable refugee groups, and examples of practical applications. These are thoughtful points that may be helpful for future work but are not essential for further revision of this manuscript.